# Adaptive Human–AI Coordination via Hierarchical Action Disentanglement

## Abstract

Human–AI collaboration requires agents that can adapt to diverse partner behaviors and skill levels while remaining robust to unseen partners. Existing methods often collapse to a single dominant behavior or learn poorly aligned skills, limiting effective coordination. We propose Intrinsic Action Disentanglement (IAD), a deep hierarchical reinforcement learning (DHRL) framework that learns distinct, partner-aware low-level action sequences conditioned on high-level latent skills. IAD introduces an intrinsic reward that explicitly encourages disentangled action distributions of the agent's low-level policy across skills, yielding an interpretable mapping between high-level decisions and partner-specific behavioral responses. By capturing temporally extended interaction patterns, IAD enables flexible adaptation to heterogeneous partner dynamics under distributional shift. We evaluate IAD in the Overcooked-AI domain across multiple layouts and diverse partner settings, including unseen simulated partners, a human-proxy model trained on human–human gameplay, and real human partners. Results show that IAD consistently outperforms strong baselines and achieves more reliable, adaptive coordination across all settings.

## 1 Introduction

Developing intelligent agents that can collaborate effectively with human partners remains a major challenge in multi-agent reinforcement learning (MARL) Klein et al. (2004); Alami et al. (2006); Bard et al. (2020). While MARL progress is often demonstrated in adversarial domains, where success is defined as outperforming an opponent Ye et al. (2020), collaboration requires agents to adapt to partners whose behaviors are diverse, unpredictable, and influenced by individual preferences and competency levels Hu et al. (2020). In real-world collaborative scenarios, human partners may demonstrate diverse coordination styles and skill levels, requiring AI agents to rapidly adjust their behavior to maintain effective collaboration. Early approaches relied on behavior cloning (BC) from human demonstrations Carroll et al. (2019), but these methods are costly and struggle to generalize beyond the training data. Deep hierarchical reinforcement learning (DHRL) Sutton et al. (1999) provides a promising alternative by decomposing tasks into reusable high-level skills and low-level actions. These skills capture common patterns of cooperative behavior and enable agents to adapt more efficiently in complex environments Eysenbach et al. (2019); Loo et al. (2023).

Despite these advances, existing DHRL skill discovery methods face key limitations in human–AI collaboration. Some approaches Eysenbach et al. (2019); Gregor et al. (2016) often produce redundant skills that capture noisy variations in observations rather than meaningful partner-relevant behavior. Other methods Loo et al. (2023) are prone to skill collapse, where multiple skills converge to similar behaviors, reducing diversity and adaptability. As a result, agents may overfit to the partners encountered during training and fail to generalize to novel partners with different skill levels or behavioral styles, limiting the effectiveness of hierarchical policies in multi-agent coordination tasks.

To address these limitations, we propose Intrinsic Action Disentanglement (IAD), a bi-level deep hierarchical reinforcement learning framework for training adaptive, partner-aware collaborative agents. The high-level skill manager observes temporally extended patterns of partner behavior and determines a skill that effectively guides the interaction. The low-level policy executes sequences of atomic actions conditioned on the

selected skill. A key contribution of IAD is a novel intrinsic reward applied to the low-level policy, which encourages each skill to produce a distinct and predictable action sequence. This reward aligns the agent's responses with partner behavioral dynamics, such as coordination style or skill level, ensuring that each skill captures a coherent mode of interaction. By disentangling skills in this way, the low-level policy learns diverse and structured action patterns, while the high-level manager dynamically selects skills that best respond to the partner's temporally extended behavior. This bi-level optimization yields a reusable skill set that generalizes robustly to previously unseen partners. Our main contributions are summarized as follows:

- We propose IAD, a novel bi-level DHRL framework for human-AI collaboration. IAD enables agents to adapt effectively to previously unseen partners characterized by diverse coordination styles and skill levels, resulting in robust coordination in collaborative tasks.

- We introduce a novel intrinsic reward for the low-level policy that explicitly encourages disentanglement of atomic action sequences across skills. This reward aligns each high-level skill with a distinct, partner-relevant behavioral pattern, preventing skill collapse and producing reusable, skill-conditioned actions that facilitate effective high-level decision-making.

- We conduct extensive evaluations in the Overcooked-AI environment across multiple layouts with diverse coordination challenges. Our experiments test generalization to previously unseen partners, including (i) large self-play partner populations with varying skill levels and coordination styles, (ii) a behavior-cloned human-proxy model trained from real human–human trajectories, and (iii) real human partners. We further analyze skill usage patterns to demonstrate that IAD produces diverse, specialized, and adaptive skill-conditioned behaviors that support effective coordination.

## 2 Related Work

Developing agents that coordinate effectively with humans has been a major focus of recent research Carroll et al. (2019); Hao et al. (2024). Carroll et al. Carroll et al. (2019) introduced the *Overcooked-AI* environment and trained PPO agents alongside human proxy models derived from human–human gameplay trajectories. Although this enhances robustness to human interaction, it relies on extensive human data, which is expensive to obtain. Hao et al. Hao et al. (2024) propose intrinsic rewards to encourage exploration of states that yield sparse extrinsic rewards when coordinating with human proxies. Fictitious Co-Play (FCP) Strouse et al. (2021) generates a pool of self-play policies and their historical versions, enabling adaptive agents without human data while capturing diverse partner behaviors. Extensions to FCP and other works further enhance heterogeneity using entropy-based objectives during training Lupu et al. (2021); Garnelo et al. (2021); Zhao et al. (2023); Loo et al. (2023), and Hidden-utility Self-Play (HSP) Yu et al. (2023) models human biases as hidden reward functions to train agents that cooperate with unseen humans. While these approaches improve partner diversity, they generally learn single-level policies, limiting their ability to reason over temporally extended behaviors critical for effective human–AI coordination.

DHRL provides a principled framework for tasks requiring temporally extended behavior Sutton et al. (1999); Flet-Berliac (2019); Pateria et al. (2021). Classical DHRL methods, including options Bacon et al. (2017); Eysenbach et al. (2019) and feudal learning Vezhnevets et al. (2017), demonstrate the benefits of temporal abstraction in single-agent settings, with recent work extending these ideas to cooperative multi-agent and human–AI scenarios Loo et al. (2023); Yang et al. (2023). Methods such as DIAYN Eysenbach et al. (2019) and related multi-agent variants Yang et al. (2023); Gregor et al. (2016) encourage behavioral diversity via intrinsic rewards that maximize mutual information between skills and states or actions. While effective in continuous control tasks Zhang et al., these approaches are less suited to multi-agent coordination Carroll et al. (2019), as they promote diversity without explicitly modeling interaction dynamics or partner-dependent behavior. Moreover, they can be overly sensitive to minor, partner-irrelevant state variations, including those induced by the agent's own actions, which may cause the high-level policy to treat trivial differences as distinct skills. As a result, the learned skills may fail to capture meaningful, partner-relevant behaviors, limiting adaptability and generalization in human–AI collaboration.

Hierarchical Population Training (HiPT) Loo et al. (2023) improves skill diversity by introducing an influence reward for shaping the high-level policy reward. However, the low-level policy in HiPT is trained solely with

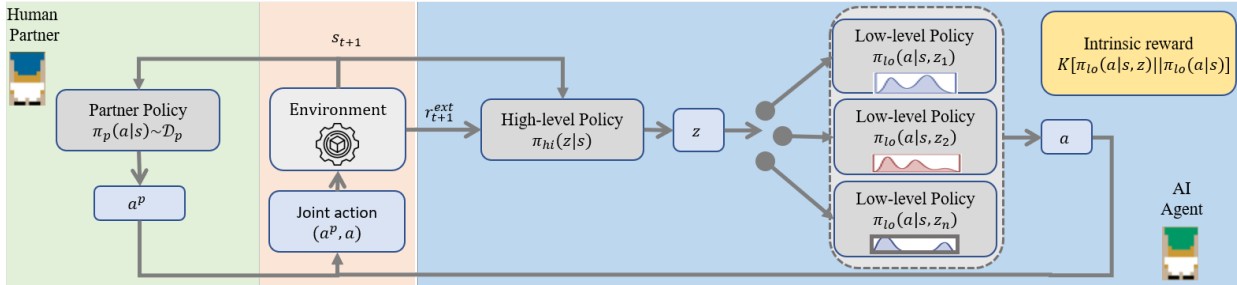

Figure 1: Overview of the IAD hierarchical framework. A partner policy $\pi^p$ is sampled per episode from $\mathcal{D}_p$. At each state $s_t$, the high-level policy $\pi_{hi}$ selects a skill $z$, with termination governed by $\beta$. The low-level policy $\pi_{lo}$ executes partner-adaptive actions conditioned on $z$. The high-level policy receives cumulative extrinsic reward over the skill duration, while the low-level policy is trained with both extrinsic and intrinsic rewards to encourage action disentanglement.

extrinsic sparse rewards and does not explicitly encode partner-dependent interaction structure, which can lead to skill collapse and limit adaptive coordination. Furthermore, both HiPT and HiPPO Li et al. randomly terminate high-level skills without proper termination criteria for low-level policies, which can limit efficient adaptation to dynamically changing partner behaviors. Overall, these limitations indicate that existing approaches either promote diversity without modeling interaction dynamics or rely on extrinsic rewards that fail to capture partner-specific coordination patterns. This underscores the need for hierarchical policies with interaction-aware intrinsic objectives. In contrast, our method introduces an intrinsic reward for the low-level policy that encourages distinct, skill-conditioned behaviors, while the high-level policy adaptively selects skills and enforces proper termination to ensure effective, partner-aware coordination.

## 3 Preliminaries

We consider a multi-agent collaborative setting in which two agents cooperate to complete shared tasks in a fully observable environment. One agent, referred to as the AI agent, is controlled by a parameterized policy $\pi_\theta(a \mid s_t)$, while the other, called the partner agent, follows a policy $\pi^p(a \mid s_t)$ sampled uniformly from a population of pretrained partners $\mathcal{D}_p$ at the start of each episode. Although the environment state is fully observable, the partner's policy is latent and fixed for the duration of an episode, inducing temporal dependencies that cannot be resolved from a single timestep. The objective is to train the AI agent $\pi_\theta$ to achieve high returns when paired with novel partner agents drawn from an evaluation population $\mathcal{D}'_p$.

The environment is modeled as a two-player Markov decision process (MDP) $\mathcal{M} = (\mathcal{S}, \mathcal{A}, \mathcal{A}^p, \mathcal{P}, r, \gamma, \rho_0)$, where $\mathcal{S}$ denotes the state space, $\mathcal{A}$ and $\mathcal{A}^p$ are the action spaces of the AI agent and the partner agent, respectively, $\mathcal{P}$ is the transition kernel, $r(s_t, a_t, a_t^p)$ is the shared team reward, $\gamma \in (0, 1)$ is the discount factor, and $\rho_0$ is the initial state distribution. At each timestep $t$, the AI agent selects an action $a_t \sim \pi_\theta(\cdot \mid s_t)$, while the partner agent executes $a_t^p \sim \pi^p(\cdot \mid s_t)$, and the next state is sampled as $s_{t+1} \sim \mathcal{P}(\cdot \mid s_t, a_t, a_t^p)$. From the AI-agent's perspective, the effective environment dynamics marginalize over the partner population:

$$\mathcal{P}_{\mathcal{D}_p}(s' \mid s, a) = \mathbb{E}_{\pi^p \sim \mathcal{D}_p} \mathbb{E}_{a^p \sim \pi^p(\cdot \mid s)} \big[ \mathcal{P}(s' \mid s, a, a^p) \big]. \tag{1}$$

For a given partner $\pi^p$, the discounted return of policy $\pi_\theta$ is given by :

$$J(\pi_\theta \mid \pi^p) = \mathbb{E}\Big[ \sum_{t=0}^{T-1} \gamma^t r(s_t, a_t, a_t^p) \Big], \tag{2}$$

and the expected return over the partner population is

$$J(\pi_\theta) = \mathbb{E}_{\pi^p \sim \mathcal{D}_p} \big[ J(\pi_\theta \mid \pi^p) \big]. \tag{3}$$

The learning objective is then

$$\theta^\star = \arg\max_\theta J(\pi_\theta), \tag{4}$$

where the expectation is taken over both partner policies and trajectory distributions.

**Deep Hierarchical Reinforcement Learning (DHRL):** Effective human–AI collaboration demands reasoning over temporally extended behaviors and adaptation to partners with heterogeneous styles and competency levels. Such partner-specific characteristics are not observable at a single timestep and must instead be learned over extended interaction horizons. Hierarchical policies facilitate temporal abstraction in reinforcement learning (RL) by enabling decision making over extended interaction horizons. This is critical in human-AI collaboration, where partner behaviors and coordination styles are only revealed through sequential interaction rather than single-step observations, even when the environment is fully observable. To capture these aspects, we adopt a hierarchical policy structure inspired by the options framework Sutton et al. (1999), consisting of a high-level manager $\pi_{hi}(z \mid s)$ and a low-level controller $\pi_{lo}(a \mid s, z)$. The high-level manager selects latent skills $z \in \mathcal{Z}$, which guide the low-level controller to produce temporally extended action sequences. Each skill $z_k$ is executed over a segment from $t_k$ to $t_{k+1} - 1$, with cumulative discounted reward given by:

$$\mathcal{R}^Z(s_{t_k}, z_k) = \sum_{t=t_k}^{t_{k+1}-1} \gamma^{t-t_k} r(s_t, a_t, a_t^p). \tag{5}$$

A stochastic termination function $\beta(z, s)$ determines when the manager selects a new skill, allowing adaptive switching in response to the collaborative context. The high-level return is the expected reward across skill segments:

$$J_{hi}(\pi_{hi}, \pi_{lo} \mid \pi^p) = \mathbb{E}\left[\sum_{k=0}^{K-1} \gamma^{t_k} \mathcal{R}^Z(s_{t_k}, z_k)\right], \tag{6}$$

while the low-level return corresponds to executing a skill within a segment:

$$J_{lo}(\pi_{lo} \mid z, \pi^p) = \mathbb{E}\left[\sum_{t=t_k}^{t_{k+1}-1} \gamma^{t-t_k} r(s_t, a_t, a_t^p)\right]. \tag{7}$$

Together, these define the hierarchical policy $\pi_\theta = (\pi_{hi}, \pi_{lo}, \beta)$. The overall learning objective is to maximize the expected return under the partner distribution:

$$\theta^\star = \arg\max_\theta \mathbb{E}_{\pi^p \sim \mathcal{D}_p}\left[J_{hi}(\pi_{hi}, \pi_{lo} \mid \pi^p)\right]. \tag{8}$$

where the inner expectation accounts for the low-level execution of skills. Optimization is performed end-to-end using proximal policy optimization (PPO) Schulman et al. (2017), enabling the agent to coordinate effectively with diverse partners across temporally extended tasks.

## 4 Method

Optimizing hierarchical policies only with extrinsic environment rewards, as defined by the high-level and low-level returns in equations (6) and (7), often results in skill collapse, where all skills converge to a dominant behavior rather than producing distinct trajectories. Extrinsic rewards are typically sparse, being received only when a subgoal or the overall task is completed. As a result, such sparse feedback provides insufficient guidance for the high-level manager to maintain meaningful diversity across skills, making it difficult to discover distinct and adaptive skill-conditioned behaviors. Furthermore, extrinsic rewards capture only task completion outcomes and fail to reflect partner-specific preferences or coordination strategies, which can vary across partners and even change dynamically within an episode. Consequently, the high-level policy cannot reliably align skill selection with the partner's style of play, leading to ineffective coordination. These limitations motivate the use of an intrinsic reward that explicitly encourages distinct low-level behaviors and

are sensitive to partner preferences, enabling the high-level manager to adapt to diverse partners. In contrast to prior intrinsic-reward and KL-regularized approaches that are largely partner-agnostic, our intrinsic objective explicitly operates at the interaction level, encouraging skill-conditioned action distributions that are sensitive to partner behavior.

### 4.1 Intrinsic Action Disentanglement (IAD)

Effective human–AI coordination requires the AI agent to adapt to partner-specific behaviors over extended interaction sequences. Each latent skill $z \in \mathcal{Z}$ should therefore correspond to a unique and reusable coordination strategy that captures the partner's coordination style over a sub-trajectory of timesteps, rather than reflecting agent-centric action patterns. At the start of each episode, a partner policy $\pi^p$ is sampled from a population of pretrained partners $\mathcal{D}_p$. Since these partner policies are pretrained and demonstrate consistent behavioral tendencies, their actions dominate the observed state transitions $s_{0:T}$ during the early phase of training, while the AI agent's policy $\pi_\theta$ is initially untrained. This ensures that the high-level manager, implemented as a recurrent network Hochreiter & Schmidhuber (1997), primarily infers latent skills $z$ from partner-driven patterns rather than from agent-centric dynamics. By leveraging these stable interaction patterns, the latent skills encode coordination strategies that align with the partner's behavior, enabling effective adaptation across diverse partners.

To ensure that each partner-driven behavior captured by the latent skill $z$ is expressed as a distinct action distribution in the AI agent, such that partner sub-trajectories are effectively mapped to corresponding sub-trajectories generated by the AI agent, we introduce Intrinsic Action Disentanglement (IAD). IAD regularizes the low-level policy $\pi_{lo}(a \mid s, z)$ to generate skill-conditioned action sequences aligned with these inferred partner behaviors. An overview of the proposed IAD framework is illustrated in Figure 1. Specifically, IAD defines an intrinsic reward that encourages the low-level policy to produce action distributions that are maximally distinct across skills:

$$I(a; z \mid s) = \mathbb{E}_{z \sim \pi_{hi}(z|s)}\Big[\mathrm{KL}\big(\pi_{lo}(a \mid s, z) \,\|\, \pi_{lo}(a \mid s)\big)\Big], \tag{9}$$

where the marginal action distribution is approximated as

$$\pi_{lo}(a \mid s) = \sum_{z \in \mathcal{Z}} \pi_{hi}(z \mid s)\, \pi_{lo}(a \mid s, z), \tag{10}$$

and $\mathrm{KL}[\cdot \,\|\, \cdot]$ denotes the Kullback–Leibler divergence. By maximizing this divergence, IAD explicitly aligns each latent skill $z$ with a sub-trajectory of actions that mirrors the partner's coordination patterns, ensuring that the high-level manager can reliably select skills corresponding to distinct partner strategies rather than transient agent-centric variations.

**Low-level objective.** Guided by the intrinsic reward defined by IAD, the low-level policy $\pi_{lo}(a \mid s, z)$ is trained to jointly maximize task performance and skill-specific diversity over a skill segment $[t_k, t_{k+1} - 1]$:

$$J_{lo}^{\mathrm{IAD}}(\pi_{lo}) = \mathbb{E}\Big[ \sum_{t=t_k}^{t_{k+1}-1} \gamma^{t-t_k} \big( r(s_t, a_t, a_t^p) + \tau\, r_{\mathrm{IAD}}(s_t, a_t, z_k) \big) \Big] \tag{11}$$

where the intrinsic reward $r_{\mathrm{IAD}}(s_t, a_t, z_k) = \mathrm{KL}\big(\pi_{lo}(\cdot \mid s_t, z_k) \,\|\, \pi_{lo}(\cdot \mid s_t)\big)$ encourages the low-level policy to produce distinct, skill-conditioned action distributions, and $\tau$ balances extrinsic and intrinsic contributions. By optimizing this objective, the low-level policy learns reusable sub-trajectories that encode partner-specific coordination patterns, providing a structured foundation for high-level skill selection.

**High-level objective.** The high-level manager $\pi_{hi}(z \mid s)$ learns to select the appropriate latent skill $z$ at each timestep to maximize the cumulative extrinsic team reward across skill segments:

$$J_{hi}(\pi_{hi}, \pi_{lo} \mid \pi^p) = \mathbb{E}\left[ \sum_{k=0}^{K-1} \gamma^{t_k} \sum_{t=t_k}^{t_{k+1}-1} r(s_t, a_t, a_t^p) \right] \tag{12}$$

Since the low-level policy is simultaneously trained to align its actions with partner-driven behaviors through the intrinsic IAD reward, the high-level manager can focus solely on selecting the most effective skill at the right time. By leveraging the temporally extended information captured in the observed state sequence $s_{0:T}$, which encodes partner dynamics via the recurrent manager, $\pi_{hi}$ adapts skill selection to the partner's coordination style. Unlike prior intrinsic-reward or KL-regularized HRL approaches Eysenbach et al. (2018), which promote skill diversity based primarily on the agent's own exploration or state distributions, IAD explicitly conditions skill learning on partner-driven sub-trajectories observed over temporally extended sequences. By ensuring that latent skills $z$ capture patterns induced by the partner's behavior, the low-level policy generates actions aligned with these partner strategies. This allows the high-level manager to leverage the temporal structure of interactions to select the most effective skill at each moment, resulting in adaptive coordination and robust multi-agent cooperation.

**End-to-End Hierarchical PPO Training:** The full hierarchical policy $\pi = (\pi_{hi}, \pi_{lo}, \beta)$ is optimized end-to-end using Proximal Policy Optimization (PPO). For each timestep $t$ within a skill segment, the low-level advantage is computed using generalized advantage estimation (GAE) Schulman et al. (2015):

$$\hat{A}_t^{lo} = \sum_{l=0}^{\infty} (\gamma\lambda)^l \delta_{t+l}^{lo}, \quad \delta_t^{lo} = \tilde{r}_t + \gamma V^{lo}(s_{t+1}) - V^{lo}(s_t), \tag{13}$$

where $\tilde{r}_t = r(s_t, a_t, a_t^p) + \lambda\, r_{\text{IAD}}(s_t, a_t, z)$, and $V^{lo}(s)$ is the low-level value estimate from the critic network. At each skill selection step $k$, the high-level advantage is computed over the cumulative extrinsic reward until termination:

$$
\begin{aligned}
\hat{A}_k^{hi} &= \sum_{l=0}^{\infty} (\gamma\lambda)^l \delta_{k+l}^{hi}, \\
\delta_k^{hi} &= \mathcal{R}^Z(s_{t_k}, z_k) + \gamma V^{hi}(s_{t_{k+1}}) - V^{hi}(s_{t_k})
\end{aligned}
\tag{14}
$$

where $V^{hi}(s)$ is the high-level value estimate. This separation ensures that low-level skills are learned based on stepwise feedback, while the high-level policy optimizes the overall effect of each skill on team performance.

The termination function $\beta(z, s) \in [0, 1]$ determines whether the current low-level skill $z$ should terminate at state $s$. $\beta(z, s)$ is optimized using the same high-level advantage signal as the manager policy. This aligns skill termination decisions with the expected cumulative reward of each skill segment, ensuring that skills terminate appropriately to maximize overall task performance. $\beta(z, s)$ is treated as a Bernoulli policy, and its loss is computed using the high-level advantage:

$$
\begin{aligned}
L^\beta(\theta_\beta) = -\mathbb{E}_k \Big[ \hat{A}_k^{hi} \log \beta(z_k, s_{t_k}) \\
+ (1 - \hat{A}_k^{hi}) \log(1 - \beta(z_k, s_{t_k})) \Big].
\end{aligned}
\tag{15}
$$

This ensures that each low-level skill executes its intended sequence fully before switching while allowing the high-level manager to adaptively terminate skills in response to partner behavior. PPO losses are computed separately for each level:

$$
\begin{aligned}
L_{lo}^{\text{PPO}}(\theta_{lo}) = \mathbb{E}_t \Big[ \min \Big( \rho_t(\theta_{lo}) \hat{A}_t^{lo}, \\
\text{clip}(\rho_t(\theta_{lo}), 1 - \epsilon, 1 + \epsilon) \hat{A}_t^{lo} \Big) \Big],
\end{aligned}
\tag{16}
$$

$$
\begin{aligned}
L_{hi}^{\text{PPO}}(\theta_{hi}) = \mathbb{E}_k \Big[ \min \Big( \rho_k(\theta_{hi}) \hat{A}_k^{hi}, \\
\text{clip}(\rho_k(\theta_{hi}), 1 - \epsilon, 1 + \epsilon) \hat{A}_k^{hi} \Big) \Big].
\end{aligned}
\tag{17}
$$

Table 1: Total mean reward (Mean ± Std) across three versions of each evaluation partner (early, intermediate, final checkpoint).

| Layout | FCP | DIAYN | HiPT | IAD |
|---|---|---|---|---|
| Cramped Room | $137.7 \pm 1.0$ | $33.8 \pm 6.4$ | $117.9 \pm 4.4$ | $\mathbf{172.96 \pm 11.85}$ |
| Asymmetric Advantages | $90.6 \pm 1.0$ | $1.5 \pm 0.7$ | $86.2 \pm 0.9$ | $\mathbf{140.63 \pm 9.57}$ |
| Coordination Ring | $83.9 \pm 5.9$ | $22.5 \pm 6.3$ | $96.0 \pm 1.3$ | $\mathbf{115.42 \pm 2.87}$ |
| Counter Circuit | $51.3 \pm 5.0$ | $1.2 \pm 1.0$ | $38.1 \pm 5.3$ | $\mathbf{60.42 \pm 0.36}$ |
| Forced Coordination | $36.7 \pm 14.4$ | $1.3 \pm 0.0$ | $35.6 \pm 13.0$ | $\mathbf{44.25 \pm 3.87}$ |

where $\rho_t(\theta_{lo}) = \frac{\pi_{\theta_{lo}}(a_t|s_t)}{\pi_{\theta_{lo}^{\text{old}}}(a_t|s_t)}$ and $\rho_k(\theta_{hi}) = \frac{\pi_{\theta_{hi}}(z_k|s_{t_k})}{\pi_{\theta_{hi}^{\text{old}}}(z_k|s_{t_k})}$ are the respective policy ratios. The joint update is then performed using the combined loss:

$$L = L_{lo}^{\text{PPO}} + L_{hi}^{\text{PPO}} + L^{\beta}, \tag{18}$$

where $L^{\beta}$ corresponds to the termination policy loss.

This training procedure ensures that the low-level policy acquires distinct, partner-aligned skills through IAD intrinsic reward, the high-level manager selects skills adaptively based on temporally extended partner behavior, and the termination function enforces proper execution of each skill segment. Together, the hierarchical policy achieves adaptive coordination and high overall team performance across heterogeneous partners. The end-to-end hierarchical PPO training, including rollout collection and policy updates of the hierarchical policy, is summarized in Algorithms 1 and 2 in the Appendix B.

## 5 Experiments

We evaluate our approach in the Overcooked-AI environment Carroll et al. (2019), a cooperative two-player benchmark adapted from the Overcooked game Games (2016). Agents collaborate to prepare and deliver soups through sequences of sub-tasks, with each successful delivery yielding a team reward of +20. The objective is to maximize cumulative team reward within a fixed episode horizon. We consider five standard layouts in Overcooked-AI, each presenting distinct coordination challenges that require adaptation to partners with diverse skills. To mitigate sparse rewards, we employ shaped rewards for intermediate sub-tasks (e.g., +3 for placing onions in the pot), which facilitate learning of partner-specific coordination strategies. Additional details on layouts are provided in the Appendix C.

### 5.1 Diverse Self-Play Partner Population

We construct a heterogeneous population of 16 self-play partners with diverse play styles and skill levels. Play-style diversity is encouraged using a negative Jensen–Shannon divergence objective Loo et al. (2023), while skill diversity is obtained by sampling policies from intermediate training checkpoints Strouse et al. (2021). During training, the agent is paired with a uniformly sampled partner from this population in each episode. For evaluation, a disjoint population generated using the same procedure is used to ensure testing on unseen partners. A visualization of partner diversity across layouts is provided in the Appendix D.

### 5.2 Baselines

We evaluate our method against several established baselines: FCP Strouse et al. (2021), DIAYN Eysenbach et al. (2019), and HiPT Loo et al. (2023). To ensure a fair comparison, all methods are trained and tested with the same heterogeneous partner populations described in the previous section. FCP trains a single-level adaptive policy directly with PPO using only the extrinsic environment reward. In contrast, DIAYN, HiPT, and IAD are hierarchical approaches optimized through the option-critic framework Sutton et al. (1999). DIAYN employs a two-stage process in which skills are first discovered using intrinsic diversity rewards, and then these skills are subsequently leveraged by the high-level policy for task performance.

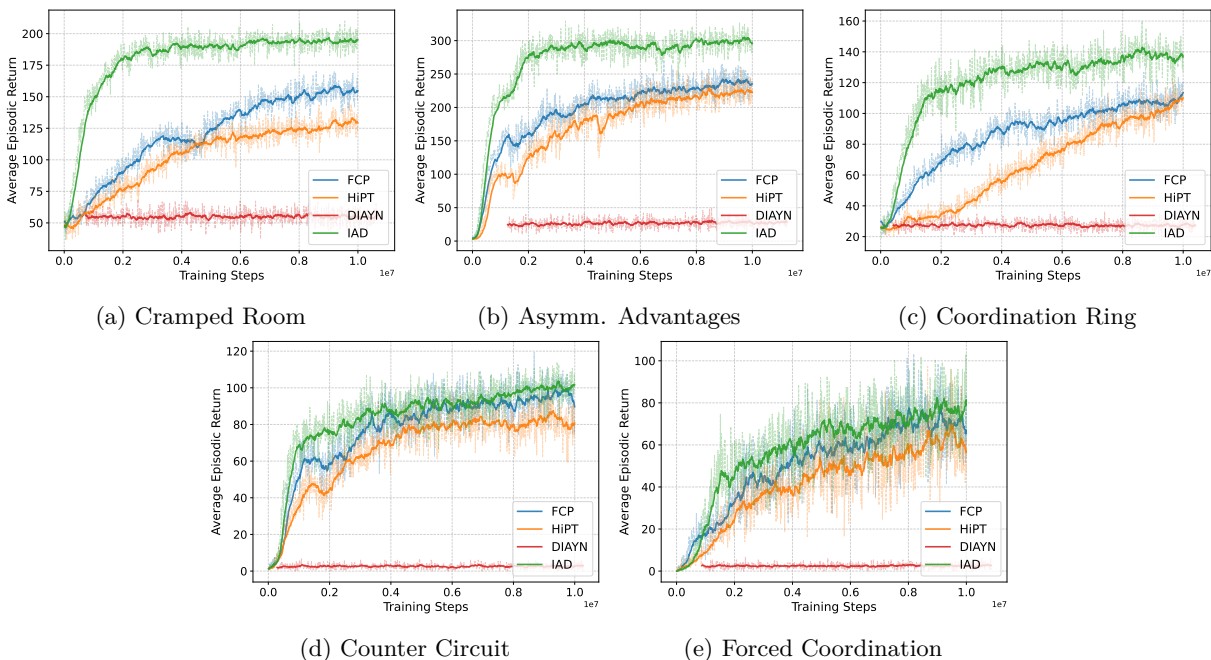

(a) Cramped Room    (b) Asymm. Advantages    (c) Coordination Ring

(d) Counter Circuit    (e) Forced Coordination

Figure 2: Average episodic return during training across 30 parallel rollout environments.

Table 2: Total mean reward (Mean ) across layouts when paired with a BC partner.

| Layout | FCP | DIAYN | HiPT | IAD |
|---|---|---|---|---|
| Cramped Room | 118.75 | 40.00 | 93.13 | **148.76** |
| Asym. Adv. | 80.00 | 0.00 | 66.25 | **124.39** |
| Coord. Ring | 79.38 | 26.25 | 77.50 | **97.35** |
| Counter Circuit | 38.13 | 1.25 | 35.00 | **43.65** |
| Forced Coord. | 30.75 | 6.30 | 25.20 | **35.43** |

HiPT and our approach train both hierarchical levels simultaneously, allowing concurrent skill discovery and task optimization.

## 5.3    Implementation Details

All methods are trained for $10^7$ environment steps with 30 parallel rollouts of horizon 400 (12,000 timesteps per update). For IAD, the intrinsic reward weight $\tau$ is annealed linearly from 1.0 to 0.05. Hierarchical policies use a backbone with three convolutional layers, two fully connected layers, and an LSTM, splitting into heads for low-level actions and value, high-level skill selection and value, and termination. The discrete skill dimension $z$ is 6 for all layouts except Forced Coordination (5). Both policies are optimized via PPO with consistent hyperparameters; some, like learning rate, vary across layouts. Value function coefficient is 0.5, PPO clipping 0.05. Full hyperparameters are in Table 5 and Table5. We provide the full implementation of IAD at https://anonymous.4open.science/r/IAD-B159/ to facilitate reproducibility.

## 5.4    Results

We evaluate our method in three partner settings. First, the agent is paired with a population of diverse self-play partners, as described in Section D. Second, we test with human-model partners trained via behavior cloning (BC) from human–human gameplay trajectories Carroll et al. (2019), providing a realistic approximation of actual human collaboration. Third, we evaluate with real human partners to directly mea-

Table 3: Mean reward of human participants paired with HiPT and IAD across Overcooked layouts.

| Layout | HiPT | IAD |
|--------|------|-----|
| Cramped Room | 80 | **118** |
| Asym. Adv. | 136 | **198** |
| Coord. Ring | 46 | **60** |
| Counter Circuit | 40 | **63** |
| Forced Coord. | 20 | **35** |

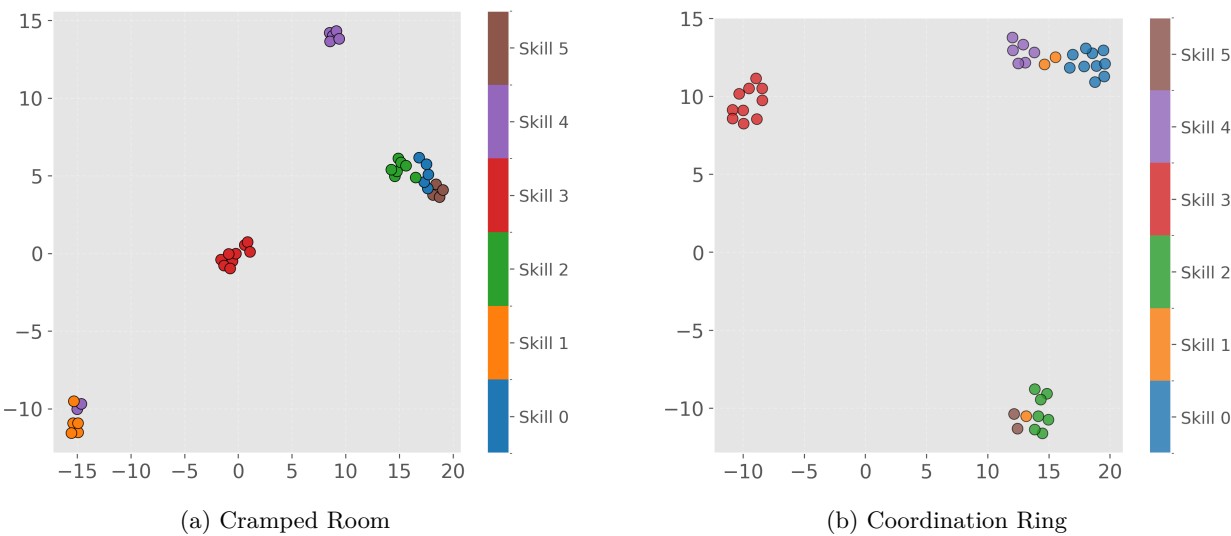

(a) Cramped Room        (b) Coordination Ring

Figure 3: 2D UMAP projections of sequence embeddings for Cramped Room and Coord. Ring layouts. Each point is a skill segment, colored by active skill. Compact skill clusters indicate intra-skill consistency, while separation reflects skill disentanglement.

sure the agent's performance and adaptability in true human–AI coordination scenarios. Across all settings, IAD consistently produces distinct skill-conditioned behaviors and achieves superior coordination and task performance compared to baseline methods.

**Evaluation with Self-Play Partner Population:** We first evaluate all methods using the heterogeneous self-play partner population introduced earlier. In our experimental setup, the population includes three types of partners. These correspond to early-stage policies, intermediate checkpoint policies, and fully trained policies. Together, they represent a range of partner skill levels, from beginner to expert, offering diverse strategies for collaboration. Such diversity has been shown to serve as a practical proxy for human behavior in collaborative tasks Strouse et al. (2021); Yu et al. (2023). For each partner, we compute the mean episodic return across both starting positions (blue and green) and across the whole evaluation partner population pool. The overall performance is then reported as the mean and standard deviation across all three partner sets. Results are summarized in Table (**??**).

Performance varies noticeably across layouts with different levels of complexity. DIAYN achieves low returns in all layouts because its skill discovery is influenced by irrelevant state variations, making it difficult to learn behaviors aligned with partner strategies. HiPT and FCP achieve moderate performance but do not perform consistently across partners due to the lack of mechanisms to structure low-level skills around partner competencies and coordination styles. In contrast, IAD explicitly disentangles low-level action distributions to align with a partner's skill level and coordination style, enabling effective adaptation and collaboration with unseen partners. This leads to the highest mean returns across all layouts, demonstrating robustness to heterogeneous and novel partners. Moreover, Figures (2a–2e) illustrate the training progress over 30 rollouts. IAD converges faster than the baselines, achieves higher returns, and maintains stable performance

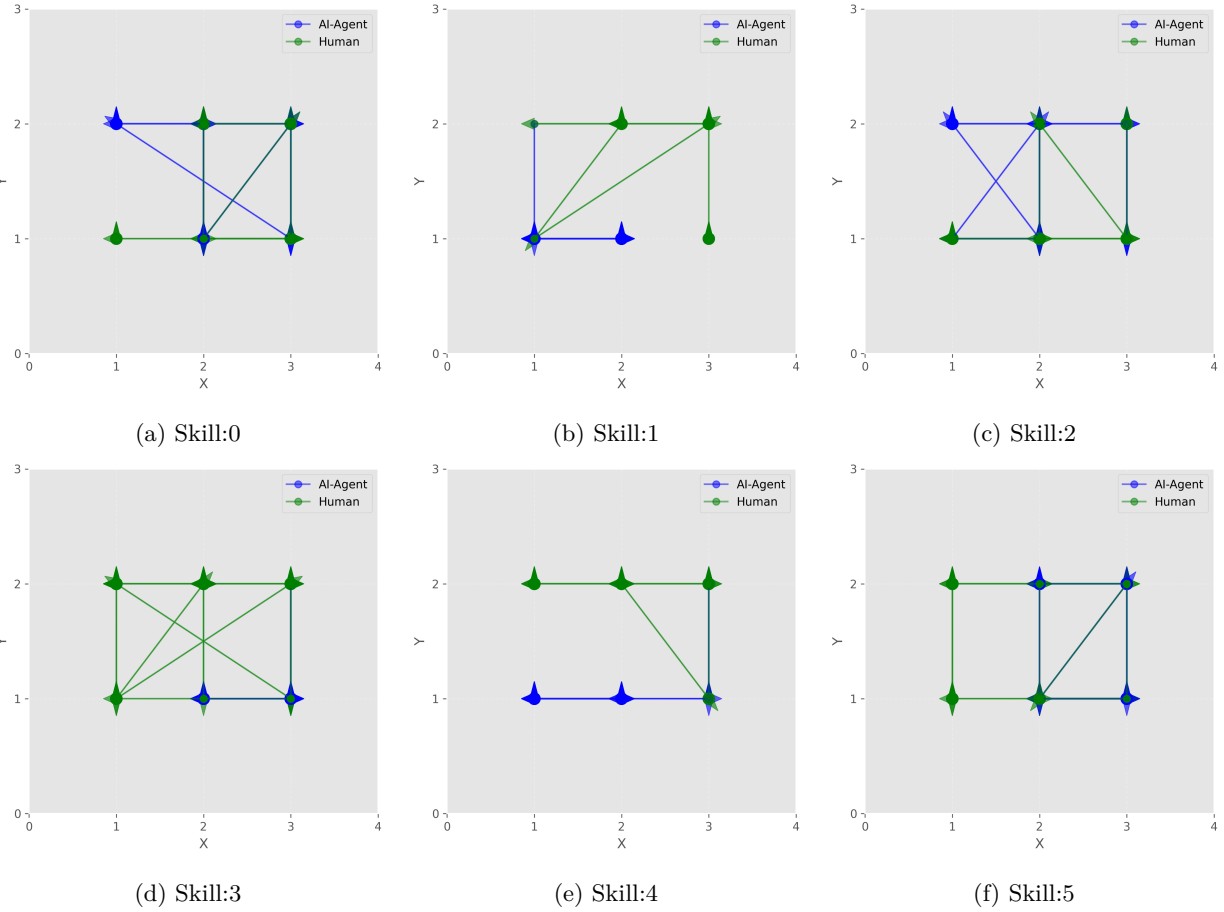

(a) Skill:0  (b) Skill:1  (c) Skill:2

(d) Skill:3  (e) Skill:4  (f) Skill:5

Figure 4: Human and AI-agent trajectories in the Cramped Room layout under different skill activations. Trajectories are reconstructed from executed movement actions (arrows), with filled and hollow circles denoting interaction and stay actions, respectively. Each skill corresponds to a distinct human sub-trajectory and a skill-conditioned AI response.

across all layout challenges, which demonstrates robust adaptation to partners with diverse skill levels and coordination styles.

**Evaluation with Human Proxy Partner:** Next, we evaluate all methods when paired with a human proxy model trained on real human–human data. This proxy is obtained via BC on publicly available human–human trajectories collected by Carroll et al. Carroll et al. (2019). The model is trained to imitate human demonstrations and then used as a fixed partner during evaluation, providing a realistic approximation of human behavior. Results are summarized in Table (2). Overall performance slightly decreases compared to evaluation with the self-play population, reflecting the limitations of BC when trained on finite human data, which can induce dominant action biases or occasional stalling in the absence of exploratory perturbations Carroll et al. (2019). IAD consistently achieves the highest returns, demonstrating its ability to generalize and adapt to previously unseen, human-like partner behaviors while maintaining effective coordination across all layouts.

**Human Subject Study: Real Human–AI Collaboration Evaluation** We evaluated IAD in real human–AI collaboration through a controlled user study, following the protocol of Carroll et al. (2019). Thirty participants were recruited via Amazon Mechanical Turk[1] , each completing two approximately 20-minute sessions: one interacting with HiPT and the other with IAD. Session order was randomized to control for ordering effects. To reduce participant fatigue and ensure manageable session lengths, only HiPT and IAD were tested, and participants who did not complete both sessions were excluded, yielding 24 valid participants. During each session, agent and human trajectories as well as rewards were recorded across all layouts. Table 3 presents the mean rewards across participants. IAD consistently achieved higher joint rewards with human partners compared to HiPT, with improvements ranging from 22% to 47%, demonstrating that skill-conditioned adaptation to partner behavior significantly enhances coordination in real-world human–AI interactions.

**Qualitative Analysis of Skill Disentanglement:** To illustrate how IAD (blue agent) adapts to human behavior, we conducted controlled sessions in the Cramped Room and Coordination Ring layouts. Each session lasted approximately 80 seconds ($\sim$482 steps), capturing diverse human gameplay. In the Cramped Room, participant demonstrated varying task execution strategies, such as picking plates from different sides or collecting soup in distinct sequences. In the Coordination Ring, participant coordinated either clockwise or counter-clockwise, requiring the AI agent to adapt its actions to align with the chosen coordination pattern. These sessions demonstrate IAD's ability to dynamically adjust its low-level behavior in response to temporally extended human strategies. Figure 3 shows two-dimensional UMAP projections of sequence-level embeddings, where each embedding represents a contiguous sub-trajectory during which a particular skill remains active. Each point is colored according to the active skill. Compact clusters for individual skills illustrate strong *intra-skill consistency*, while separation between clusters indicates clear *inter-skill disentanglement*. The visualization demonstrates that embeddings corresponding to the same skill form compact, well-separated clusters, reflecting both the temporal consistency and behavioral disentanglement enforced by the intrinsic reward. Once a high-level skill is selected in response to a human behavior pattern, the low-level policy produces coherent action sequences conditioned on that skill. These sequences adapt to the human behavior that triggered the skill, demonstrating that IAD aligns high-level skill selection with human intent while maintaining consistent, skill-specific action execution.

To further examine the behavioral dynamics captured by IAD, Figure 4 shows that each high-level skill consistently aligns with a distinct pattern of human behavior. In response, the AI agent produces coherent, skill-conditioned action sequences that adapt to the triggering human trajectory. This demonstrates that IAD aligns skill selection with human intent while generating predictable, partner-aware behaviors.

## 6  Conclusion

This work addressed the challenge of effective Human–AI coordination in complex environments, where agents must adapt to diverse and dynamically changing partner behaviors. We proposed IAD, a hierarchical

---

[1]https://www.mturk.com/

framework that enables adaptive coordination by learning distinct, skill-conditioned low-level behaviors through an intrinsic reward mechanism. By mapping each high-level skill to a unique low-level action pattern, IAD captured variations in partner strategies and skill levels, allowing robust adaptation to diverse coordination styles. We evaluated IAD in the Overcooked domain across multiple layouts and against a large, unseen population of partners with varying behaviors, including a human-proxy model trained from human–human data. Across all settings, IAD consistently outperformed baseline methods, achieving higher returns and more reliable coordination. Further analysis of skill usage, including entropy and switching behavior, showed that IAD dynamically adapted its skills to partner diversity, enabling generalization to novel partners.

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

---

**Algorithm 1** IAD Rollout

---

0: **Input:** Partner population $\mathcal{D}_p$, high-level policy $\pi_{hi}$, low-level policy $\pi_{lo}$, termination $\beta$
0: **Parameters:** Intrinsic weight $\tau$, rollout horizon $T$
0: Initialize buffers $\mathcal{B}_{hi}, \mathcal{B}_{lo}$
0: **for** each parallel rollout **do**
0:     Sample partner $\pi^p \sim \mathcal{D}_p$
0:     Reset environment $s_0 \sim \rho_0$, $t \leftarrow 0$
0:     **while** $t < T$ **do**
0:        Sample skill $z_t \sim \pi_{hi}(z|s_t)$
0:        **repeat**
0:           Sample low-level action $a_t \sim \pi_{lo}(a|s_t, z_t)$
0:           Partner action $a_t^p \sim \pi^p(a|s_t)$
0:           Step: $s_{t+1}, r_t \leftarrow \text{EnvStep}(s_t, a_t, a_t^p)$
0:           Compute intrinsic reward $r_{\text{IAD}}(s_t, a_t, z_t)$
0:           Store $(s_t, z_t, a_t, r_t + \tau r_{\text{IAD}}, s_{t+1})$ in $\mathcal{B}_{lo}$
0:           Sample termination $b_t \sim \beta(z_t, s_{t+1})$
0:           $t \leftarrow t + 1$
0:        **until** termination $b_t$ or $t \geq T$
0:        Compute segment return $R^Z = \sum_{t_k}^{t_{k+1}-1} r_t$ and store $(s_{t_k}, z_k, R^Z)$ in $\mathcal{B}_{hi}$
0:     **end while**
0: **end for**
0: **Return:** Buffers $\mathcal{B}_{lo}, \mathcal{B}_{hi}$ =0

---

## A    Appendix

## B    IAD Pseudocode.

The pseudocode for the proposed Intrinsic Action Disentanglement (IAD) framework is provided in Algorithms 1 and 2. Algorithm 1 describes the rollout procedure, where a partner policy $\pi^p$ is sampled from the pretrained population $\mathcal{D}_p$, the high-level manager selects latent skills $z_t$, and the low-level policy executes skill-conditioned actions while receiving intrinsic rewards $r_{\text{IAD}}$ that encourage alignment with partner behavior. Skill termination decisions are determined by $\beta(z_t, s_{t+1})$, and trajectories are stored in separate buffers for high- and low-level policy updates.

Algorithm 2 outlines the end-to-end policy update procedure using Proximal Policy Optimization (PPO). Low-level and high-level advantages are computed via Generalized Advantage Estimation (GAE), and the termination function is updated to ensure proper execution of skill segments. PPO losses for the low-level policy, high-level manager, and termination function are computed and used for joint updates of the hierarchical policy. These algorithms provide the implementation details necessary to reproduce our training procedure.

## C    Environment

We evaluate our approach in the Overcooked-AI environment Carroll et al. (2019), a cooperative two-player benchmark adapted from the Overcooked game Games (2016). In the Overcooked environment, agents collaborate to prepare and deliver soups by completing a sequence of sub-tasks, including picking onions, placing them in the pot, waiting for cooking, and serving the soups. Each successful soup delivery yields a team reward of +20, and the objective is to maximize the cumulative team reward within a fixed episode horizon.

We evaluate five standard layouts in Overcooked-AI, which include Cramped Room, Asymmetric Advantages, Coordination Ring, Counter Circuit, and Forced Coordination (Figure 5). Each layout presents unique coordination challenges that evaluate agents' ability to adapt to partners with diverse skills and play styles.

---

**Algorithm 2** Policy Update

0: **Input:** Buffers $\mathcal{B}_{lo}, \mathcal{B}_{hi}$; low-level policy $\pi_{lo}$, high-level policy $\pi_{hi}$, termination policy $\beta$
0: **Parameters:** PPO clip $\epsilon$, discount $\gamma$, GAE $\lambda_{GAE}$, epochs $E$, minibatch size $M$
0: Compute low-level advantages $\hat{A}_t^{lo}$ using equation: (13):
0: Compute high-level advantages $\hat{A}_k^{hi}$ using equation: (14):
0: Termination advantages: $\hat{A}_t^{\beta} = \hat{A}_k^{hi}$
0: **for** epoch $e = 1$ to $E$ **do**
0:    **for** each minibatch of size $M$ from $\mathcal{B}_{lo}, \mathcal{B}_{hi}$ **do**
0:       Compute probability ratios: $\rho_t^l, \rho_k^h$ and $\rho_t^{\beta}$
0:       Compute PPO losses using equation: (18)
0:       Update parameters $\pi = (\pi_{lo}, \pi_{hi}, \beta)$
0:    **end for**
0: **end for**
0: **Return:** Updated policies $\pi_{lo}, \pi_{hi}, \beta = 0$

---

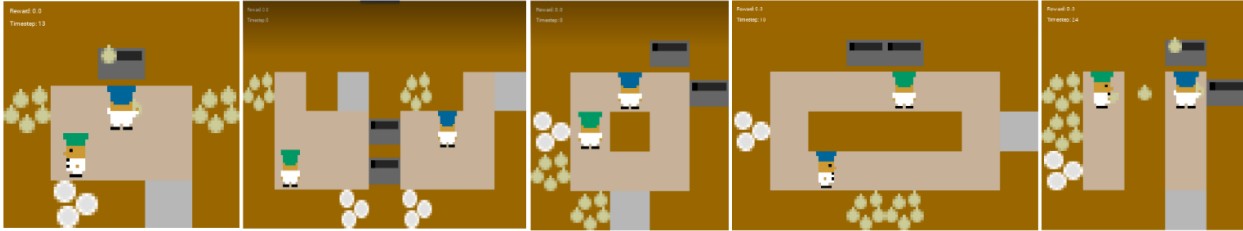

Figure 5: The five standard Overcooked layouts (left to right): Cramped Room, Asymmetric Advantages, Coordination Ring, Counter Circuit and Forced Coordination.

In Cramped Room, tight spaces increase the likelihood of collisions, requiring agents to coordinate turn-taking and navigate carefully. Asymmetric Advantages involves partners specializing in complementary roles, demanding flexible skill selection to achieve coordinated behaviors. Coordination Ring enforces a looped workflow, emphasizing alignment with partners' directional preferences, while Counter Circuit focuses on precise timing and item exchanges due to counter-mediated interactions. Forced Coordination imposes physical separation, highlighting the need for sequenced cooperation and dynamic task routines. These layouts serve as a challenging testbed for evaluating human–AI coordination and the agent's ability to adapt to diverse partner behaviors.

In addition to the extrinsic reward of +20 per successful soup delivery, we employ shaped rewards to provide intermediate feedback on critical sub-tasks, such as picking up or placing an onion in the pot (+3 reward). Without such intermediate signals, the extrinsic reward is sparse and offers limited guidance, making it difficult for the agent to learn how to coordinate effectively with partners. Shaped rewards help the agent identify partner-specific sub-task preferences and explore action sequences that contribute meaningfully toward the overall task goal, rather than arbitrary actions.

## D  Diverse Self-Play Partner Population

To train agents that coordinate effectively with partners characterized by diverse styles and skill levels, we construct a heterogeneous population of 16 self-play partners. Each partner differs in play style and skill level, providing diverse interaction experiences during training. Play style diversity is encouraged using a negative Jensen–Shannon Divergence during training Loo et al. (2023), while partners of varying skill levels are included by sampling agents from intermediate training checkpoints Strouse et al. (2021). During training, the AI agent interacts with a partner sampled uniformly from this population in each episode, exposing it to a broad range of coordination behaviors. For evaluation, a separate disjoint population of the same size is generated using the same procedure, ensuring that the agent is tested on novel partners not seen during training. Figure (6) visualizes the structure of these partner populations, showing clusters of

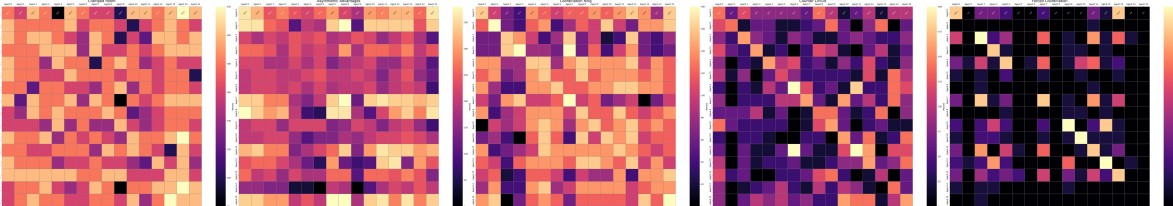

(a) Cramped Room  (b) Asymmetric Adv. (c) Coordination Ring  (d) Counter Circuit  (e) Forced Coord.

Figure 6: Heatmaps showing the average pairwise returns of a diverse partner population in Overcooked across five layouts. Returns are averaged over five episodes of length $T = 400$. Lighter squares indicate pairs of partners with similar play-styles, while darker squares indicate less compatible partners. The heatmaps reveal distinct clusters of partner play-styles in Forced Coordination and Counter Circuit, while Cramped Room, Asymmetric Advantages, and Coordination Ring reflect more uniform behavior, likely due to simpler coordination requirements.

Table 4: General hyperparameters used across all layouts.

| Hyperparameter | Value |
|---|---|
| Entropy coefficient | $0.01 \rightarrow 0$ (linear decay) |
| Value function coefficient | 0.5 |
| Clipping coefficient | 0.05 |
| Optimizer | Adam |
| Discount factor $\gamma$ | 0.99 |
| GAE parameter $\lambda_{GAE}$ | 0.98 |
| Batch size | 64 per environment |
| Parallel environments | 30 |

similar play-styles and the heterogeneity across different layouts, which the agent must adapt to for robust coordination.

## E  Implementation Details

All methods are trained for $10^7$ environment steps using 30 parallel rollouts, each with a horizon of 400 timesteps, yielding 12,000 timesteps per update. For IAD, the intrinsic reward weight $\tau$ is linearly annealed from 1.0 to 0.05 during training. The hierarchical policies use a backbone network with three convolutional layers, two fully connected layers, and an LSTM layer. The network branches into separate heads for low-level actions and value prediction, high-level skill selection and value estimation, and a termination head. The discrete skill dimension $z$ is set to 6 for all layouts, except for Forced Coordination, where it is 5.

Table 5: Layout-specific training parameters. Learning rates decay linearly from the initial value to the initial value divided by the decay ratio over training.

| Layout | Initial Learning Rate | Decay Ratio |
|---|---|---|
| Cramped Room | $1.0 \times 10^{-3}$ | 3 |
| Asymmetric Advantages | $1.0 \times 10^{-3}$ | 3 |
| Coordination Ring | $6.0 \times 10^{-4}$ | 1.5 |
| Forced Coordination | $8.0 \times 10^{-4}$ | 2 |
| Counter Circuit | $8.0 \times 10^{-4}$ | 3 |

Both the low-level and high-level policies are optimized using Proximal Policy Optimization (PPO) with consistent hyperparameters across layouts, while some parameters (e.g., learning rate) are adjusted per layout to account for specific coordination challenges. The value function coefficient is fixed at 0.5, and the PPO clipping parameter is set to 0.05. A full list of hyperparameters, including general and layout-specific settings, is provided in Table (4) and Table (5).

