# OpenReview forum: "Adaptive Human–AI Coordination via Hierarchical Action Disentanglement"
_TMLR — Under review for TMLR_

### Review · Reviewer_snnb · 2026-05-31

**Summary Of Contributions:**

The paper introduces Intrinsic Action Disentanglement (IAD), a hierarchical reinforcement learning framework for human–AI collaboration. The core idea is to learn high-level latent skills and low-level action policies while explicitly encouraging different skills to produce distinct action distributions through a KL-based intrinsic reward. The authors argue that this prevents skill collapse and enables adaptation to partners with different coordination styles and skill levels.

Strengths：
1. Addresses an important problem: robust adaptation to heterogeneous human partners.
2. Proposes a simple and intuitively motivated intrinsic reward based on action-distribution disentanglement.
3. Evaluates across multiple partner settings, including real human participants.
4. Demonstrates consistent empirical improvements over several relevant baselines.

Weaknesses：
1. The theoretical justification for why action-distribution disentanglement specifically captures partner-dependent coordination patterns is limited.
2. Several claims regarding partner-awareness and alignment with human intent are stronger than what is directly demonstrated experimentally.
3. The human study is relatively small (24 valid participants) and only compares against HiPT rather than all baselines.

**Audience:**

Yes

**Audience Explanation:**

The proposed intrinsic reward mechanism represents a potentially useful contribution to hierarchical skill learning and partner adaptation. Researchers working on cooperative MARL, human-aware AI systems, zero-shot coordination, and hierarchical RL would likely find the findings interesting, particularly because the method is evaluated against established baselines and includes both proxy-human and real-human experiments.

**Claims And Evidence:**

Yes

**Claims Explanation:**

The empirical evidence generally supports the paper's primary claim that IAD improves coordination performance compared to the selected baselines in the Overcooked domain. Across all five layouts, IAD achieves higher average returns when paired with unseen self-play partners, behavior-cloned human proxies, and real human participants. The consistency of these gains across multiple evaluation settings strengthens the empirical case. The qualitative analyses also provide some evidence that the learned skills are more distinct and structured than those produced by prior methods. The UMAP visualizations and trajectory examples suggest that different latent skills correspond to different behavioral patterns.

However, some stronger claims are not fully supported by the presented evidence. In particular, the paper repeatedly argues that the intrinsic reward causes skills to become aligned with partner-specific behaviors and human intent. While the observed performance improvements are consistent with this interpretation, the experiments do not conclusively establish this causal relationship. Additionally, the human-subject evaluation includes only a comparison against HiPT and uses a relatively modest sample size.

**Requested Changes:**

1. Strengthen evidence for partner-awareness claims, e.g., (1) provide experiments demonstrating that learned skills genuinely correspond to partner-dependent behaviors rather than merely diverse action modes; (2) include analyses showing how skill selection changes across different partner types.

2. Improve the human-subject evaluation.

3. Address theoretical concerns. Better justify why maximizing skill-conditioned action divergence should lead to partner-relevant coordination strategies.

---

### Review · Reviewer_Ly8i · 2026-06-05

**Summary Of Contributions:**

Summary

This paper proposes a policy training method for human-AI collaboration tasks. To train the model to adapt human policy, the author adopts hierarchical reinforcement learning method with intrinsic reward to train diverse skills. The proposed method demonstrates better empirical performance than existing methods among several collaboration setting including human collaborator.


Strength

The author proposes a method for human-AI collaboration task that empirical demonstrates good performance.

The author conducts experiment on several setting including human-AI collaboration study.

The author also conducts visualization of the learned skill to demonstrate their diversity.


Weakness

The proposed method is the inclusion of entropy-like subskill policy reward to hierarchical reinforcement learning. I think the technical novelty is small. Further, the motivation is the skill adaptation to the observed partner policy. However, it seems this adaptation mechanism is not implanted nor experimentally evaluated.

Figure 3 looks like some skill embeddings are overlapping. Further, since the policy of the existing methods are not visualized, based on this figure alone, it cannot be definitively stated that the proposed method acquires diverse policies.

**Audience:**

Yes

**Audience Explanation:**

The proposed method demonstrates better score than existing methods. The author conducts real human-AI collaboration experiment. These results may have interest to the researcher of this area.

**Broader Impact Concerns:**

Since the paper proposes a model for human-AI collaboration task, it may be better to discuss the possible negative effect to human partner caused by the proposed model.

**Claims And Evidence:**

No

**Claims Explanation:**

As written in weakness, the proposed method does not reflect the motivation to adaptively select the skill depending on the collaborator’s policy. Further, this adaptiveness is not experimentally evaluated.

Further, the skill diversity is not compared against existing methods.

**Requested Changes:**

Clarification of novelty.

Discussion that the proposed method adaptively selects skills based on the collaborator’s skills and experimental verification. e.g. plot of sampled collaborator’s policy and selected skill vector z.

Comparison of skill diversity to existing methods.

Fix some broken references e.g. HiPPO paper does not have year information in page 3. There is ?? in page 9.

---

### Review · Reviewer_ogKZ · 2026-06-12

**Summary Of Contributions:**

## **Strengths**

**(S1. readability)** This work is well written and the flow is easily readable.
**(S2. motivation)** This work is well motivated, and hierarchical adaptation is intuitive and reasonable for human-AI coordination. The idea of pushing for action disentanglement is neat.
**(S3. results)** The reported empirical results are strong and consistent across unseen self-play partners, BC partners, and real human participants. IAD outperforms the reported baselines across five Overcooked layouts.

---

## **Weaknesses**

**(W1. Claim 1)** Perhaps the most important one. From what I can gather, I think IAD intrinsic reward enforce skill-aware disentanglement, but not **direct** partner-aware disentanglement, which is the central claim of the paper. The proposed IAD encourages z-conditioned low-level action distributions to be distinguishable, but it does not directly enforce that z captures partner behaviour or partner trajectory-level coordination. The partner-aware part comes from I believe the inherent diversity of the partner population and the recurrent high-level policy of partner policies. This is also related to (Q2) and (M3).

**(W2. Claim 2)** This work uses temporal information through the recurrent high-level policy and through the skill until termination. However, the proposed IAD is still a per-timestep action-distribution term, not a **sequence-level** disentanglement term. Therefore, the paper’s claim that IAD explicitly disentangles action sequences is stronger than what the objective **directly** enforces.

**(W3. Previous Works Results not Matching)** As far as I can tell, the partner policies are trained with the same procedure as in HiPT, and the evaluation procedure is the same. If the training/evaluation procedure is intended to match HiPT, the performance of HiPT and FCP appears inconsistent with the original HiPT results, and the difference is quite significant. Furthermore, in HiPT's original work, HiPT outperforms FCP slightly in all five tasks, while in this work it's reversed. The authors should clarify the reason, and if there are differences in the procedure/hyperparams/etc. It is possible that I've misunderstood something about the experiments settings.

**(W4. Ablation Study)** An ablation study of proposed method **without IAD** is missing, which would would show the effectiveness of the proposed reward scheme that encourages skill-conditioned action diversity is the source of performance gains. Another ablation, perhaps following HiPT's termination strategy, would also clarify how the proposed termination mechanism is important.

---

## **Questions**

**(Q1. Bernoulli policy)** The manuscript claims that "HiPT and HiPPO terminate high-level skills without proper termination criteria for low-level policies", can the authors expand on this more? How does this differ concretely from the termination mechanism in HiPT? An ablation study on this would be nice. Related to (W3).

**(Q2. Diversity of partners)** I'm wondering if there are any ways to show the diversity of the partners' play styles, and show how this diversity is essential to the performance of the proposed method. Although Figure 6 visualises pairwise returns among partners, it is still unclear how diverse the partner population is in terms of behaviour or strategy, and how essential this diversity is to IAD’s performance.

**(Q3. Metrics)** How is "total mean reward" calculated? Is the the sum of mean reward of the 3 partner versions over rollout seeds? What about their individual std across different rollouts? In this sense, please also report the std of Table 2 and Table 3. Mainly, I wish to understand the differences between this work's reported results with the HiPT's reported results.

**(Q4. Ablations)** How is skill dimensions z set? How is this value, i.e. 5 or 6, chosen for all tasks? Furthermore, why set to 5 for Forced Coordination task? How does this skill dimension affect performance?

**(Q5. Visualisation)** Related to Q4, in the visualisation of Figure 3, it seems that some of the skills are quite close to each other. Does that offer some insights that maybe the skill dimensions can be set differently? Related to (Q2) and (M3), because I think there are no specific term to enforce "skill-diversity" for the high-level policy in the proposed method.

---

## **Minor Comments**

**(M1. Presentation)** Should change the "0:" in the Algorithm 1 and Algorithm 2, such as  line numbers are increasing.

**(M2. Typos)** This is a non-exhaustive list in no particular order:
	- There's a extra space in the captions of table 2.
	- "Table 5" and Table5" in [5.3 Implementation Details]. spacing not consistent.
	- repeated same citation in Evaluation with Human Proxy Partner: Carroll et al. Carroll et al. (2019).
	- The dashes are not consistent, e.g. towards the end of paragraph of human subject study, human-AI vs real-world have different dashes.
	- reference style are not consistent; e.g. Figure (2) vs Table 5 vs Table (2) vs Figure 3
	- In page 9, there's an incorrectly rendered cref. I think this is referring to Table 1?

**(M3. Implicit Assumption)** I think there's a assumption behind the whole framework that there exist diverse-enough, large number of partner policies for training. Technically not a weakness, but makes me wonder if the proposed method would work if there's only a couple of (2? 4?) partner policies, especially when they are not diverse enough. Related to Q2 and W1.

**Audience:**

Yes

**Audience Explanation:**

This work address an interesting problem of human-AI collaboration where the policy is trained to adapt to different skill-levels of human for the given task. The proposed method can be easily adapted to related problems.

**Broader Impact Concerns:**

There are no particular broader impact concerns as far as I'm concerned.

**Claims And Evidence:**

Yes

**Claims Explanation:**

Partly yes, because the major claims are mostly supported indirectly. The first claim for partner-aware disentanglement are support indirectly because the contribution objective IAD only promotes skill-aware disentanglement, and the second claim of sequence-level disentanglement is also partially support because IAD is calculated as a per-timestep term. Both of these issues can be addressed either by tuning down the claims, or supporting evidence.

**Requested Changes:**

**(R1. Clarification)** Address (M1) and (M2) by fixing the presentation and typo issues. It's possible that I've missed some.

**(R2. Clarification)** Address (Q3) by clarifying how quantitative results are calculated.

**(R3. Clarification)** Address (W3) by expanding on the procedures of training/evaluation/hyperparameter mismatches if there is compared to previous works. Address (Q1) by explaining more in detail too.

**(R4. Claim about partner-awareness)** Address (W1) and (Q2), perhaps by tuning down the claim, or clarify how partner-awareness is obtained in the framework. Would also address (M3).

**(R5. Claim about sequences)** Address (W2), perhaps by tuning down the claim.

**(R6. Ablations)** Address (W4) and (Q4) by adding ablation studies on the effectiveness of IAD. If possible, add an ablation study on importance of selecting skill dimension, which would address (Q5). If possible, add an ablation studying on how different these termination strategy affect results, which would address (Q1).